# Improving Surface Wear Resistance of Polyimide by Inserting KH550 Grafted GO

**DOI:** 10.3390/polym15112577

**Published:** 2023-06-04

**Authors:** Chen Liu, Jingfu Song, Gai Zhao, Qingjun Ding

**Affiliations:** 1State Key Laboratory of Mechanics and Control for Aerospace Structures, Nanjing University of Aeronautics and Astronautics, Nanjing 210016, China; liuchen39@nuaa.edu.cn (C.L.); dingqingjun@nuaa.edu.cn (Q.D.); 2School of Naval Architecture & Ocean Engineering, Jiangsu Maritime Institute, Nanjing 211170, China; songjingfu1984@126.com

**Keywords:** polyimide, molecular dynamics simulation, tribology, graphene oxide

## Abstract

To improve the wear resistance of polyimide (PI), surface modification was developed. In this study, the tribological properties of graphene (GN), graphene oxide (GO), and KH550-grafted graphene oxide (K5-GO) modified PI were evaluated by molecular dynamics (MD) at the atomic level. The findings indicated that the addition of nanomaterials can significantly enhance the friction performance of PI. The friction coefficient of PI composites decreased from 0.253 to 0.232, 0.136, and 0.079 after coating GN, GO and K5-GO, respectively. Among them, the K5-GO/PI exhibited the best surface wear resistance. Importantly, the mechanism behind the modification of PI was thoroughly revealed by observing the wear state, analyzing the changes of interfacial interactions, interfacial temperature, and relative concentration.

## 1. Introduction

Ultrasonic motors are electric motors that operate based on the principle of ultrasonic waves, converting the ultrasonic vibration of the stator into the motion of the rotor through frictional force. Researchers have also developed various types of ultrasonic motors to meet the needs of different application fields, such as surface vibration wave type motors [1,2], standing wave type motors [3], and resonant type motors [4]. Ultrasonic motors are characterized by high speed, high precision, and high efficiency [5], and have been widely researched and applied both domestically and internationally. Friction materials play a crucial role in ultrasonic motors, directly impacting the performance and lifespan of the motors. In countries like the United States, Japan, Germany, and others, the field of ultrasonic motors holds a leading position. Researchers in these countries are dedicated to the theoretical research and engineering applications of ultrasonic motors. They focus on issues such as high-speed driving, high-precision control, and energy efficiency of ultrasonic motors. Uchino et al. conducted a study on compact rotary ultrasonic motors with a diameter of 1.5 mm and a torque of only 0.1 mN·m [6]. This research provides valuable references for the further application and development of ultrasonic motors and opens up new possibilities for designing smaller and higher-performance ultrasonic motors. The Chang’e-3 lunar probe, as China’s lunar exploration mission, also utilized ultrasonic motor technology. In the Chang’e-3 mission, lightweight, high efficiency, and high reliability were essential requirements. The small size, low weight, and high efficiency of ultrasonic motors made them an ideal choice. The motor played a crucial role in spacecraft adjustment, navigation systems, and control of the lunar probe. Its high speed and high precision ensured precise operation and stability of the probe, while its high efficiency characteristics improved energy utilization efficiency. In addition, ultrasonic motors have shown great potential in the field of medical devices [7]. With advancements in medical technology, there is an increasing demand for minimally invasive surgery and precise treatments. Ultrasonic motors, with their small size and high precision control, have become an ideal choice for minimally invasive surgical instruments and precise positioning devices. Researchers are exploring the applications of ultrasonic motors in areas such as endoscopy, robot-assisted surgery, and tissue sample collection, bringing revolutionary changes to the medical field. It should be noted that despite the remarkable achievements in ultrasonic motor research, there are still challenges and issues that need to be addressed. For example, thermal management, energy efficiency, and long-term stability of ultrasonic motors require further investigation. Additionally, the manufacturing cost and reliability of ultrasonic motors are also important factors to consider. Polyimide and its composites, as important friction materials, have advantages such as high mechanical strength, good stability, and excellent electrical insulation [8], so they are widely used in industrial production and daily life [9,10,11,12,13]. In the aerospace field, polyimides with good radiation resistance are ideal materials for ultrasonic motors [14]. However, the wear resistance of pure PI is relatively weak, which limits its wider application in the aerospace field. Therefore, many scholars are committed to improving the wear resistance, radiation resistance, and high-temperature resistance of polyimides to expand their range of use and service life.

The study conducted by Li et al. demonstrates that polyimide materials exhibit 51.7% higher efficiency compared to traditional polytetrafluoroethylene materials, making them excellent friction materials for ultrasonic motors [15]. The research explores the potential application of polyimide in ultrasonic motors by modifying its structure and formulation. The researchers discovered that treating polyimide with nanosecond pulsed fiber lasers to create micro-pits can alter the material’s friction characteristics and wear resistance. Experimental results indicate that optimized formulations of polyimide materials exhibit good friction performance and long-term stability in ultrasonic motors. An overseas study proposed a nanofiller-based reinforcement strategy to improve the friction performance of polyimide [16]. Researchers added nanoscale graphene sheets as fillers to the polyimide matrix, forming a composite material. Experimental results showed that the polyimide composite with added nanoscale graphene sheets exhibited lower friction coefficient and excellent wear resistance in ultrasonic motors. This study provides a new approach for enhancing the performance of polyimide friction materials through the use of nanofillers. Many researchers have found that the wear resistance of polymer composites can be improved by filling modification agents, surface modification of polyimide molecular structure, surface coatings, and other methods [17,18,19,20,21,22]. Breki et al. improved the friction performance of polyimide-based self-lubricating nanocomposites by using gas-phase synthesized tungsten diselenide (WSe_2_) nanoparticles as reinforcements. The results showed that the tribological properties of the nanoparticle-reinforced composite were significantly improved, and both the friction coefficient and adhesion coefficient decreased [23]. Cai et al. studied the effect of nanoscale silicon dioxide (SiO_2_) on the frictional performance of polyimide-based composites, and found that nanoscale SiO_2_ can significantly improve the wear and abrasion resistance of PI-based composites [24]. Currently, adding various fibers to the matrix is one of the most effective methods, such as carbon fibers, glass fibers, and aramid fibers, etc. Zhang et al. have studied the effect of short carbon fiber filling on polyimide and found that the addition of short carbon fibers significantly improves the tribological properties of the composite material [25]. Panin et al. also found that chopped carbon fiber (CCF) can enhance the mechanical properties and tribological properties of PI composites [26]. Wu et al. also studied the effect of carbon fiber length on enhancing polyimide-based materials and found that polyimide with 100 μm carbon fiber exhibited lower friction coefficient and wear rate than other materials. Additionally, as the carbon fiber length increased, the interfacial bonding strength between the carbon fiber and matrix deteriorated [27]. Li et al. also used homogenous copper oxide (CuO) nanowires to solve the problem of weak interfacial bonding between carbon fiber and polyimide. The results showed that CuO nanowires can effectively improve the interfacial compatibility with polyimide, and enhance the mechanical and tribological properties of the composite material [28]. Cai et al. have studied the effect of carbon nanotubes (CNTs) on the frictional properties of polyimide (PI). The results showed that CNT/PI composites had lower friction coefficient and wear rate compared to pure PI [29]. Chen et al. have also systematically studied the effects of several carbon systems, including graphite, carbon fiber, and carbon nanotubes, on the tribological properties of polyimide composites under seawater lubrication conditions. The results showed that the addition of any filler could improve the wear resistance of polyimide under seawater lubrication [30]. Graphene and graphene oxide are two-dimensional materials that have received considerable attention in recent years, and their addition in appropriate amounts can improve the mechanical properties of materials. Sekiguchi et al. investigated the effect of adding polytetrafluoroethylene (PTFE), graphite (Gra), and molybdenum sulfide (MoS_2_) on pure polyimide (PI). The results showed that the PI composites containing PTFE and Gra had lower friction coefficients and wear rates than pure PI [31]. Song et al. also investigated the effect of coupling agent treatment on glass fiber (GF) to improve the wear resistance of polyimide. The wear mechanism was analyzed using SEM. The study showed that the mechanical and wear properties of the composite filled with surface-treated GF were improved, and the treatment effect of KH550-Aminopropyltriethoxysilane (KH550) was the best [32]. From this, it can be seen that the introduction of KH550 grafted graphene oxide enhances the surface wear resistance of polyimide. The research conducted by Shan et al. demonstrated that the surface modification with KH550 improved the dispersion uniformity of nano Al_2_O_3_ in anhydrous ethanol system [33]. Li et al. also found that after adding KH550, the tensile fracture of the waste paper fiber/polylactic acid composite was smoother, and the interfacial compatibility between waste paper fiber and polylactic acid was greatly improved. When KH550 solution with 5% mass fraction was added, the tensile strength and flexural strength of the waste paper fiber/polylactic acid composite added with KH550 increased by 11.2% and 8.4% respectively compared with those without KH550 [34]. Material Studio is capable of reducing the expensive laboratory costs, minimizing experimental time, and providing relatively accurate predictions of the mechanical properties of materials. Therefore, it has been widely applied in computational materials science. It offers a variety of simulation and modeling tools for studying the structure, properties, and reactions of materials.

This study aims to investigate the mechanism of graphene, graphene oxide, and KH550-grafted graphene oxide in improving the wear resistance of polyimide through molecular dynamics simulations. The friction and wear properties of polyimide are compared by MD simulation to analyze their performance from a microscopic perspective.

## 2. Model Establishment and Simulation Process

The modeling and simulation in this paper were conducted using the molecular dynamics software Material Studio 2019. Firstly, the initial structure of polyimide was determined, as shown in Figure 1a. Subsequently, three different structures were constructed: graphene, graphene oxide, and KH550-grafted graphene oxide. These structures had dimensions of 42.3 Å × 42.3 Å and were illustrated in Figure 1b, Figure 1c, and Figure 1d, respectively. The thickness of graphene oxide was 6 Å, while the KH550-grafted graphene oxide (K5-GO) had a thickness of 13 Å. In the K5-GO structure, six silane coupling agent molecules are grafted onto the surface of graphene oxide. In the X-direction, the distance between two silane coupling agents was 20 Å, while in the Y-direction, the distance between each silane coupling agent molecule was 14 Å. The composite need dynamic optimization before calculation. During the simulation process, each silane coupling agent molecule keep stationary compare with location in the PI matrix.

A cubic lattice with dimensions of 50 × 50 × 50 Å^3^ was firstly constructed, and then the modifier was placed at the top of a repeating unit box and fixed in place. Finally, using Monte Carlo rules, PI polymer chains with a degree of polymerization of 2 were filled into the box according to the actual density of 1.3 g/cm^3^, as shown in Figure 2. Figure 2a was pure PI model, while Figure 2b–d were PI composites modified by GN, GO, and K5-GO, respectively. To facilitate identification, GN, GO, and K5-GO were colored green.

In order to achieve a more reasonable structure, geometric optimization was performed on the original model. The detailed parameters for the optimization [35] process are provided in Table 1. First, geometric optimization was performed using the Smart method with an energy convergence criterion of 2 × 10^−4^ kcal/mol and an iteration number of 1 × 10^4^. Various parameters were adjusted during the geometric optimization process to refine the molecular arrangement and improve the overall stability of the system. The optimization process involved iterative adjustments of bond lengths, bond angles, and torsional angles, aiming to minimize the potential energy of the system. Subsequently, to further relax the molecular chains of the model, an annealing process was conducted using the NVT ensemble for 15 picoseconds. The annealing temperature was gradually increased from 300 K to 600 K in three increments, with a subsequent geometric optimization of 2000 steps after each heating to ensure a reasonable model structure. During the annealing process, a temperature schedule was employed to gradually increase and then decrease the temperature. This allowed the structure to undergo thermal fluctuations and relax into a more favorable conformation. Each cycle of annealing involved a series of molecular dynamics simulations, where the system was subjected to simulated heating and cooling. The choice of time step, integration method, and temperature range was carefully considered to strike a balance between computational efficiency and accuracy. After the annealing process, the obtained structures were evaluated based on their total energies. The structure with the lowest total energy was considered the most energetically favorable and selected for further analysis. The selected structure was then subjected to dynamic equilibrium simulations, allowing it to evolve over an extended period to ensure the elimination of any remaining stresses or instabilities. By employing this approach, the optimization and annealing procedures aimed to refine the structure, eliminate residual stress, and achieve a stable and energetically favorable configuration. These steps were crucial in obtaining a reliable model for further analysis and understanding of the system’s properties and behavior. All optimization processes were conducted using the Condensed-Phase Optimized Potentials for Atomistic Simulation Studies (COMPASSII) force field based on Ewald and atom-based methods for analyzing intermolecular interactions.

During the operation of ultrasonic motors, the friction material mainly experiences two types of loads: the preloading force of the system and the shear force generated by friction. Therefore, when designing friction materials, it is important to consider their ability to withstand both loading and shear, which refers to the mechanical properties of the materials. In this study, molecular dynamics simulations were performed to conduct 12 tensile tests and calculate the Young’s modulus and shear modulus of PI composites. In the tensile tests, a strain rate of 0.01 was applied along the *X*-axis while maintaining constant volume along the other axes.

To investigate the rules and action mechanisms of GN, GO, and K5-GO on the frictional properties of PI composites, 50.6 × 50.6 × 7.2 Å^3^ Cu atomic layers were built on both the top and bottom for frictional pairs and substrate, as shown in Figure 3. During the friction process, a normal pressure of 10 MPa and a relative sliding speed of 0.1 Å/ps were applied to the upper Cu layer, and the simulation time was 600 ps. The temperature and time step of the friction process were set to 300 K and 1 fs, respectively. Through the output trajectory files, a detailed analysis of the frictional properties of PI and its composite materials can be conducted. These trajectory files record key information such as the position, velocity, and interactions of the materials during the friction process. This data can be utilized to study the influence of different materials on frictional performance and the mechanisms of their interactions.

## 3. Results and Discussion

### 3.1. Mechanical Properties of PI Composites

The mechanical performance of the composites were shown in Figure 4.The Young’s modulus of pure PI was 3.54 GPa, and the addition of GN and GO increased the Young’s modulus of the composite material by 18% and 72%, respectively. The addition of K5-GO increased the Young’s modulus of the composite material to 6.87 GPa, which was a 94% increase compared to pure PI. The trend of the shear modulus was similar to that of the Young’s modulus, with the K5-GO/PI composite having the highest shear modulus of 1.8 GPa, which was 1.16 times that of pure PI. The shear modulus of GN/PI and GO/PI also increased by 3% and 8%, respectively, compared to pure PI. Previous studies have suggested [36] that the interaction between the modifier and PI can contribute to the improvement of mechanical properties of the PI composite. Therefore, the interaction energy between the modifier and PI was extracted, as shown in Figure 5. The larger the interaction energy, the stronger adsorption effective between the modifier and the PI matrix, thus improving the mechanical properties of the composite. Among them, the interaction energy between K5-GO and PI was the largest with around 1050 kcal/mol, because the silane coupling agent grafted on the surface of GO can form a mechanical interlocking effect with PI molecular.

### 3.2. Tribological Properties of PI Composites

Firstly, the friction coefficient of pure PI and PI composites was calculated as a function of friction process time, as shown in Figure 6. It can be clearly seen that the friction coefficient begins to stabilize after approximately 180 ps. The average friction coefficient of pure PI in the stable stage is 0.253, while that of the GN/PI composite is 0.232, a decrease of 8% compared to pure PI. The friction coefficients of GO/PI and K5-GO/PI composites are 0.136 and 0.079, respectively, which have a decrease of 46% and 68% compared to pure PI. This result indicated that GN and GO had better lubrication which can reduce the interaction between polyimide molecules and Cu, resulting in a decrease in the friction coefficient. In addition, the silicon coupling agent-grafted modified GO with strong adsorption effect for PI molecular reduced the frictional force with Cu, which was consistent with mechanical performance improvement, shear modulus variations. It is also consistent with previous experimental results [37,38,39].

In addition, the dynamic evolution of the PI composites during friction process was also observed, the model exhibits periodic variation along the *X*-axis, but during the friction simulation, we compare the portion of the copper layer in contact with the 50 Å polyimide, as shown in the Figure 7. Under the same frictional conditions, pure PI had the largest shear deformation, while the addition of GN, GO, and K5-GO resulted in relatively smaller deformations. The K5-GO/PI composite shows the smallest deformation. Because GN and GO with high mechanical strength and excellent wear resistance can withstand larger external loads and shear stress, and thus protect the PI matrix. Moreover, K5-GO have strong mechanical interlocking force with PI, which have the best wear resistance. K5-GO/PI with the weakest effect with Cu have the smallest shear deformation as shown in Figure 7d. This further verified the improvement effect of K5-GO on the mechanical and tribological properties of PI matrix.

In order to deeply explore the wear mechanisms, the interaction energy between the friction pair and the PI composites were calculated. The smaller the interaction energy between the Cu layer and the PI composite, the smaller the adsorption force between them, making it easy shear with lower friction force. Figure 8 shows the interaction energy between the Cu layer and PI composites. It can be clearly seen that interaction energy between PI matrix and Cu layer decreased from 2850 to 2800 kcal/mol after filling GN and GO. It continuously reduced to about 2770 kcal/mol after inserting K5-GO. Among them, the interaction energy between K5-GO/PI and the Cu layer is the smallest. The occurrence of this phenomenon is due to the van der Waals adsorption between GN and PI, which weakens their interaction with Cu. In contrast, GO exhibits stronger adsorption ability due to its internal hydrogen bonding structure. Moreover, the addition of K5-GO, which possesses a mechanical interlocking structure, enhances the adsorption effect. Therefore, the interaction between the modified PI and Cu is weakened, which is consistent with the results of the friction experiments mentioned earlier.

Then, to further reveal the mechanism of the effect of the modifier addition on friction and wear, the variation of temperature and relative atomic concentration during the friction process was analyzed. Firstly, the temperature distribution of the polymer material in the thickness direction was extracted, as shown in Figure 9. From the temperature distribution of Pure PI, it can be seen that the highest temperature of 350 K is obtained near the contact of the friction pair. The addition of the modifying agents GN, GO, and K5-GO reduced the temperature of the composite material by 13%, 14%, and 15%, respectively. According to the theory proposed by Hu [40], friction is actually a process of energy conversion. As can be seen from the variation of friction coefficient in Figure 7, the higher the friction coefficient, the higher the temperature. So, the pure PI shows the highest interface temperature 350 K, while K5-GO/PI have the lowest temperature with 299 K.

In addition to temperature analysis, a study was conducted on the relative atomic concentration during the friction process to further understand the effects of modifier addition. Relative atomic concentration refers to the distribution of different elements within the material. By analyzing the changes in atomic concentration, we can gain insights into the redistribution of atoms and their impact on material properties. The relative concentration along the Z direction of the PI composites was extracted, as shown in Figure 10. It can be seen from the relative concentration curve that the relative concentration of Pure PI is higher at both sides than the middle place because of boundary effect, reaching 3.65 and 3.67, respectively. So, the interaction effect between PI and the friction pair was analyzed in Figure 8, which had a strong effect on the progress of friction. After adding GN, GO, and K5-GO, the relative concentration of the PI composite exhibits a peak value around 40 Å. This peak is attributed to the accumulation of carbon atoms on the top surface of the PI composites, resulting in a significantly higher concentration compared to the middle region.

## 4. Conclusions

In this study, the reinforcement effect of graphene, graphene oxide, and KH550-grafted graphene oxide on the polyimide composites was investigated using molecular dynamics simulations. The main results were as follows:Adding graphene based nanomaterials can improve the mechanical properties of PI composites, in which K5-GO exhibits the best effect, increasing the Young’s modulus to 6.87 GPa and the shear modulus to 1.8 GPa, which are 94% and 16% higher than pure PI. Besides, the enhancement mechanism was revealed by analyzing the interaction energy between the modifiers and PI molecular. The higher the interaction energy, the stronger the adsorption effect, and resulting in the the better the mechanical performance.Compared with pure PI, the surface modified PI composites have smaller friction coefficients, among which K5-GO can reduce the friction coefficient to one-third of the original value. The K5-GO/PI also shows the smallest shear deformation. By analyzing the interfacial interaction energy between PI and the Cu layer, it was found that the interfacial interaction between PI and the friction pair would decrease after the addition of modifiers, further confirming the enhanced effect of modifiers on the PI matrix. These results are helpful for a deeper understanding of the structure and properties of polymer-based composites, and provide important references for the design and manufacture of high-performance polymer composites.

## Figures and Tables

**Figure 1 polymers-15-02577-f001:**
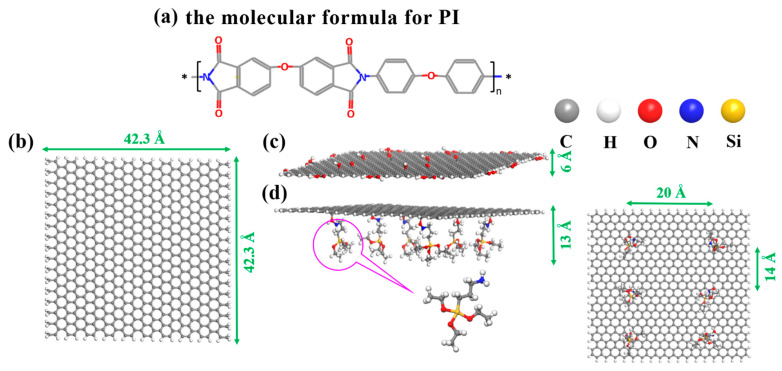
The molecular formula of (**a**) PI, (**b**) GN, (**c**) GO, (**d**) K5-GO.

**Figure 2 polymers-15-02577-f002:**
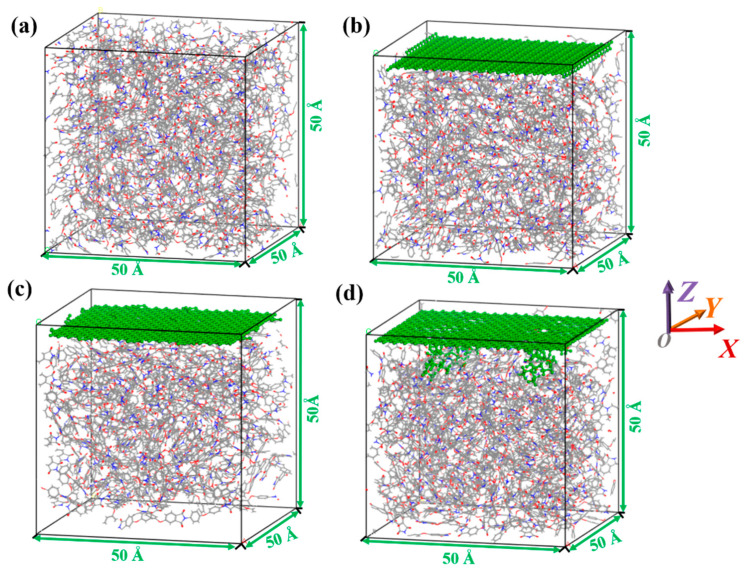
The periodical cell of (**a**) PI, (**b**) GN/PI, (**c**) GO/PI, (**d**) K5-GO/PI.

**Figure 3 polymers-15-02577-f003:**
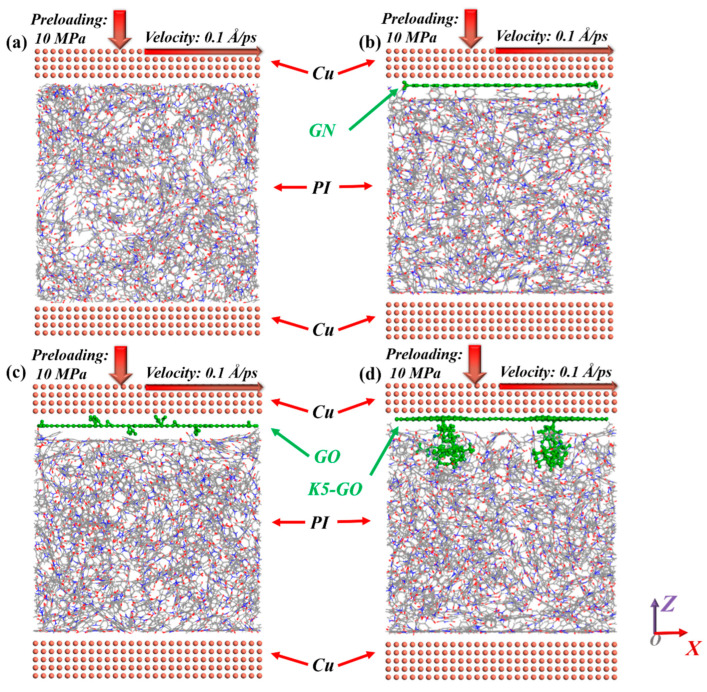
The friction models of (**a**) PI, (**b**) GN/PI, (**c**) GO/PI, (**d**) K5-GO/PI composites sliding against Cu.

**Figure 4 polymers-15-02577-f004:**
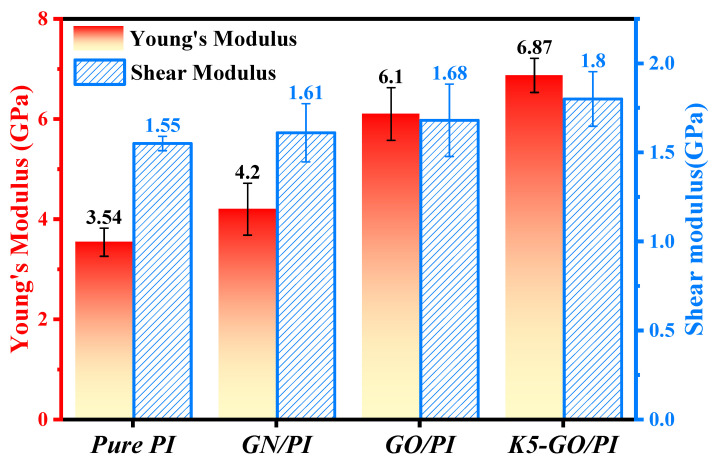
The Young’s modulus and shear modulus of the PI composites.

**Figure 5 polymers-15-02577-f005:**
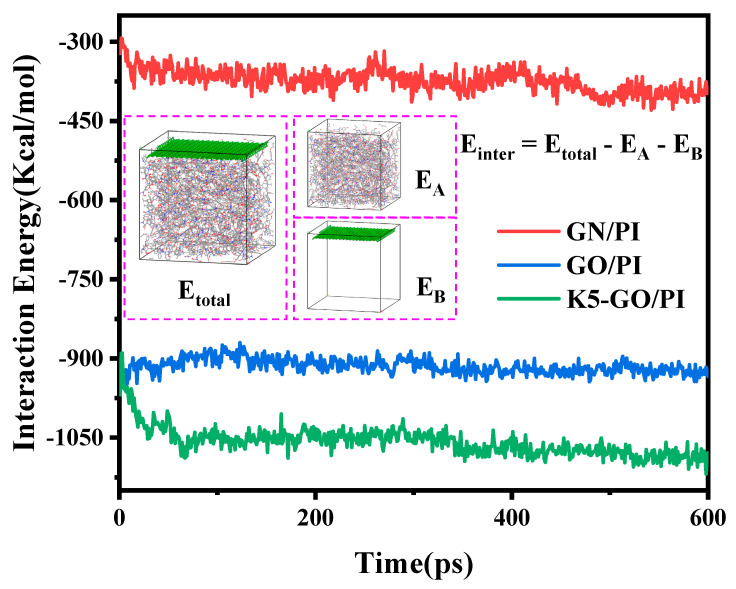
The interaction energy between the modifier and PI.

**Figure 6 polymers-15-02577-f006:**
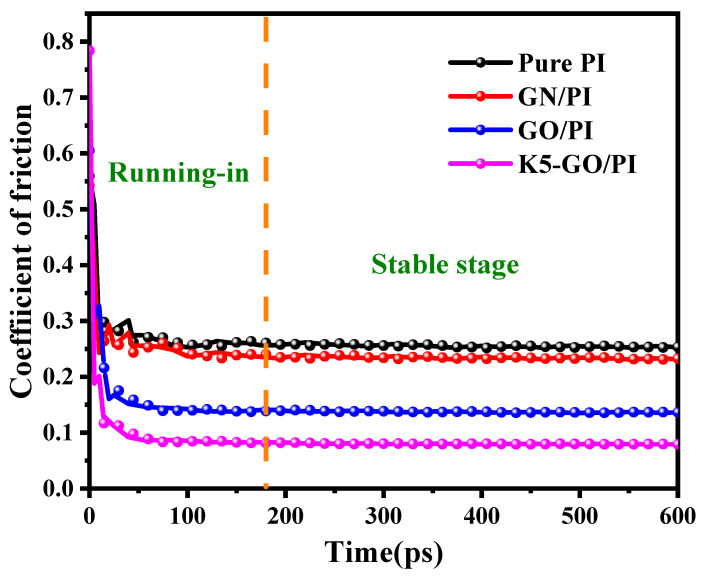
Friction coefficient variations of PI composites with time.

**Figure 7 polymers-15-02577-f007:**
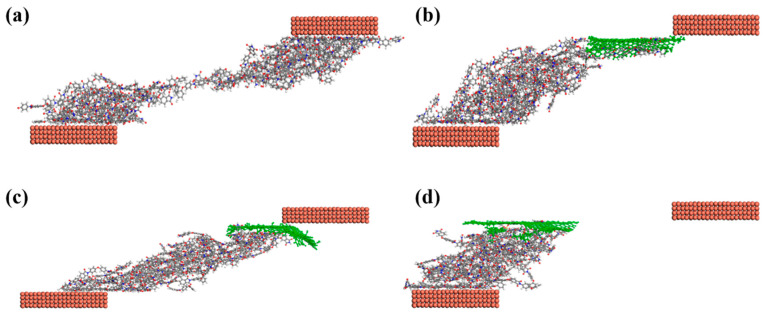
Shear deformation of (**a**) Pure PI, (**b**) GN/PI, (**c**) GO/PI, (**d**) K5-GO/PI.

**Figure 8 polymers-15-02577-f008:**
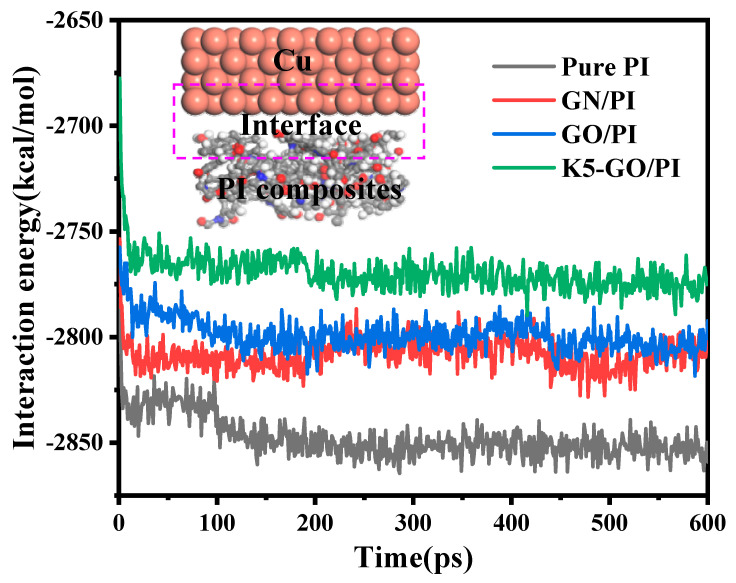
The interaction energy between PI composite and the Cu layer.

**Figure 9 polymers-15-02577-f009:**
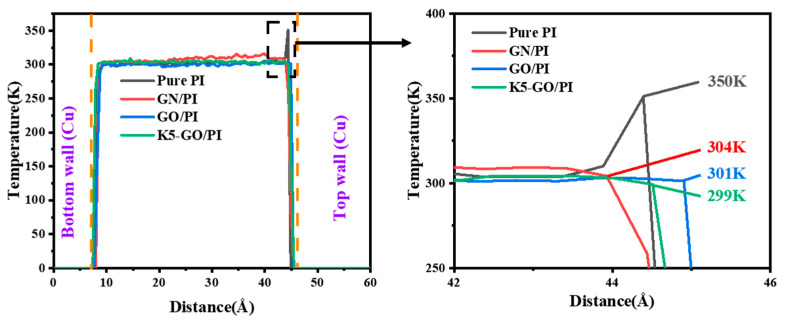
Temperature profiles of PI composites in the thickness direction.

**Figure 10 polymers-15-02577-f010:**
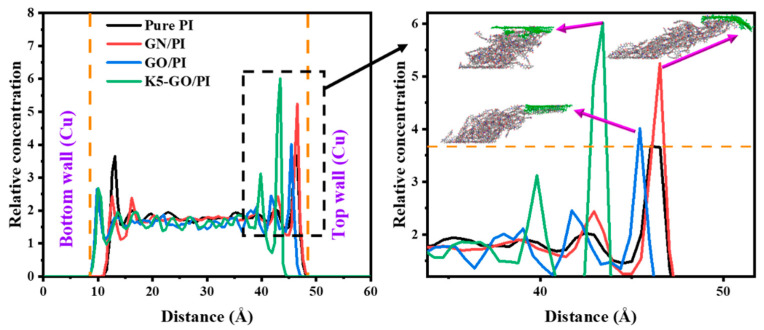
Relative concentration of PI composites in the thickness direction.

**Table 1 polymers-15-02577-t001:** Optimization process.

Optimization Process	Algorithm	Convergence Criterion	Temperature	Time
Geometry optimization	Smart	1 × 10^−4^ kcal/mol 0.005 kcal/mol/Å	/	
Anneal	Nose thermostat	/	300–600 K	
NVT	Nose thermostat	/	300 K	300 ps
NPT	Berendsen barostat	/	300 K	600 ps

## Data Availability

No new data were created, all data is available in Section 3.

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
