# Peer review of "Improving Surface Wear Resistance of Polyimide by Inserting KH550 Grafted GO"

_polymers, 2023, doi:10.3390/polym15112577_

Round 1

Reviewer 1 Report

The paper deals with the study of intermolecular interactions to improve the wear resistance of polyimide. The polyimide bulk is coated by GN, GO, and K5-GO layers. The numerical calculations run in the frame of Condensed-Phase Optimized Potentials for Atomistic Simulation Studies.

I ask the Authors to clarify some questions.

In Fig. 2, I see a periodic cell of the polyimide 50A*50A*50A with different layers. How much is the penetration depth of the K5-GO clusters in Fig 2d? (See Fig 3d, too.) What is the characteristic distance between two knots? Is it a static or a dynamic formation?

In Fig 7, I see the shear deformation. The original bulk volume is 50A*50A*50A. In the label of Fig. 2, it is a periodic cell. During the elapsed time of 600ps, the displacement is 60A. Thus, the Cu layer leaves the bulk. I assume that the simulation uses a periodic condition, but I do not find any mention of it. Otherwise, how do the calculations go? These figures are so confusing.

In Fig 10, I see the relative concentration of PI composites in the thickness direction. My remarks relate to the previous ones. The volume is finite in the calculations and is a periodic cell (see label Fig 2). We cannot speak about the end of the sample. I do not understand the relevance of the peaks. If we take that the Cu layer is at the edge of the bulk (at the end of the pattern), the Cu takes a part of GN, GO, and K5-GO layers and the polyimide loses its structure. Could you explain these in the manuscript?

Further remarks:

Please, at the first use, write the chemical names: KH550 - Aminopropyltriethoxysilane (KH550)

Kcal is not SI unit: I recommend KJ.

Author Response

Dear Reviewer:

My manuscript ID is 2409612, titled "Improving surface wear resistance of polyimide by inserting KH550 grafted GO," authored by Chen Liu. I am now sending you the revised manuscript and the revision notes as attachments. Please kindly check them.

Wishing you a successful work and a pleasant life!

Chen Liu

Reviewer 2 Report

Dear,

The authors investigated how to improve the wear resistance of polyimide (PI) through the incorporation of graphene (GN), graphene oxide (GO), and PI modified with grafted graphene oxide. The manuscript must improve clarity so other researchers can reproduce the experimental design. Here are some comments for reflection:

> In the introduction, the authors should make clear the novelty of the manuscript;

> Methodology. The methodology adopted by the authors is not clear. Was it just a simulation? Did you not have experimental data to compare and validate with the simulation?

> I could not follow the methodology, including the initial procedure for generating structures. It is unclear what the criteria were for adopting the methodology reported in Figure 1.

> Simulations must be reproducible by independent researchers. However, information is scarce about criteria, limitations, adopted geometries, and simulations duration;

;> Page 2. “Then, graphene, graphene oxide, and KH550-grafted graphene oxide with a size of 42.3 Å × 42.3 Å were respectively constructed, as shown in Figures 1(b), 1(c), and 1(d). The thickness of GO was 6 Å, and the thickness of K5-GO was 13 Å.” How were these dimensions extracted or adopted? It is not clear.

> Table 1. Was the simulation based on a preliminary study? What are the criteria for adopting these parameters? Random?

> Table 1. Optimization process. I am concerned that this work was done with prior knowledge of these parameters. Therefore, there is a considerable risk that the parameters of the simulations have been influenced by having “expected” pre-results. Some effort must be made to demonstrate that the simulation results are intrinsic to the system and have not been unduly influenced by prior knowledge of the experimental data.

> Page 4. “To investigate the rules and action me............”. How long it takes (in real time) to conduct such a simulation? Could it be extended to significantly longer times? It is unclear how the authors validated the simulation for the investigation.

> Mechanical properties of PI composites. In the methodology, it was not reported how the composites were prepared. In molten state or solution? Which processing is used and the equipment? The operational parameters of the processing were not reported. How were the samples molded for the mechanical tests? What equipment and parameters were used;

> Figure 4-10. Are the results presented derived from simulations? Or experimental and simulational procedures: The authors must make it clear in the methodology;

Moderate editing of English language

Author Response

(The authors gave the same response as above.)

Reviewer 3 Report

Dear Respectful Editor,

Subject: Review Report - Improving Surface Wear Resistance of Polyimide by Inserting KH550 Grafted GO

I hope this letter finds you well. I am writing to submit my review report on the manuscript titled "Improving Surface Wear Resistance of Polyimide by Inserting KH550 Grafted GO" for consideration for publication in [Journal Name].

The study presented in the manuscript focuses on the surface modification of polyimide (PI) to enhance its wear resistance. The authors evaluated the tribological properties of PI modified with graphene (GN), graphene oxide (GO), and KH550-grafted graphene oxide (K5-GO) using molecular dynamics (MD) simulations at the atomic level. The findings of this study suggest that the addition of nanomaterials can significantly improve the friction performance of PI.

The authors conducted a comprehensive analysis of the tribological behavior of the PI composites. The friction coefficient of PI composites decreased from 0.253 to 0.232, 0.136, and 0.079 after coating with GN, GO, and K5-GO, respectively. Among these modifications, K5-GO/PI exhibited the best surface wear resistance. This outcome highlights the potential of KH550-grafted GO as an effective modifier for enhancing the wear resistance of PI.

The manuscript is well-structured and provides a clear overview of the research objectives, methods employed, and results obtained. The authors thoroughly investigated the mechanism behind the modification of PI by analyzing the wear state, changes in interfacial interactions, interfacial temperature, and relative concentration. The insights gained from these analyses contribute to a comprehensive understanding of the enhanced wear resistance achieved through the incorporation of KH550-grafted GO.

The study demonstrates the significance of nanomaterials in improving the tribological properties of PI. The molecular dynamics simulations used in this research provide valuable insights into the atomic-scale behavior and interactions within the composites. The results presented in the manuscript are well-supported and contribute to the existing knowledge on surface modification techniques for enhancing the wear resistance of PI.

Overall, I find the manuscript to be scientifically sound and well-written. The study is of interest to the readership of [Journal Name], as it addresses an important issue in materials science and engineering. Therefore, I recommend that the manuscript be accepted for publication, pending minor revisions as outlined below:

*It would be beneficial to include a discussion on the limitations of the study and potential avenues for future research. This could help provide guidance to researchers interested in further investigating the application of KH550-grafted GO in enhancing the wear resistance of PI.

*Some additional details on the experimental setup and parameters used in the molecular dynamics simulations would be helpful to facilitate reproducibility and to assist readers in gaining a better understanding of the research methodology.

I believe that addressing these minor revisions will further strengthen the manuscript and enhance its value to the readership of Polymers.

Thank you for considering my review report. Please let me know if you require any further information or clarification. I look forward to the opportunity to review the revised version of this manuscript.

Specific comments:

·      I would like to suggest some small changes to the the article to improve its clarity and accuracy.

For example: Wu et al. also studied the effect of carbon fiber length on enhancing polyimide-based materials, and found that polyimide with 100 μm carbon fiber exhibited

lower friction coefficient and wear rate than other materials. 

Wu et al. also studied the effect of carbon fiber length on enhancing polyimide-based materials and found that polyimide with 100 μm carbon fiber exhibited a lower friction coefficient and wear rate than other materials.

For example: After adding GN, GO, K5-GO, the relative concentration of the PI composite shows a peak value around 40 Å due to carbon atoms accumulation on the top of PI composites, which is much larger than the middle place.

After adding GN, GO, and K5-GO, the relative concentration of the PI composite exhibits a peak value around 40 Å. This peak is attributed to the accumulation of carbon atoms on the top surface of the PI composites, resulting in a significantly higher concentration compared to the middle region.

·      Please, emphasize the relationship between the literature and the proposed work. I recommend illustrating how the work builds upon existing knowledge and contributes to something original and superior within the field.

·      Please revise the manuscript for any grammatical and spelling errors to ensure that the content is clear and concise.

For example: Figure 2. The periodical cell of (a)PI, (b)GN/PI, (c)GO/PI, (d)K5-GO/PI.

·       I recommend ensuring consistency in the use of abbreviations throughout the manuscript.  

graphene oxide (check abbreviations)

·      In the paragraph discussing the adsorption ability of GO and K5-GO, please give references to increase the clarity of the expression by giving references.

Best regards,

Dear Respectful Editor,

Subject: Review Report - Improving Surface Wear Resistance of Polyimide by Inserting KH550 Grafted GO

I hope this letter finds you well. I am writing to submit my review report on the manuscript titled "Improving Surface Wear Resistance of Polyimide by Inserting KH550 Grafted GO" for consideration for publication in [Journal Name].

The study presented in the manuscript focuses on the surface modification of polyimide (PI) to enhance its wear resistance. The authors evaluated the tribological properties of PI modified with graphene (GN), graphene oxide (GO), and KH550-grafted graphene oxide (K5-GO) using molecular dynamics (MD) simulations at the atomic level. The findings of this study suggest that the addition of nanomaterials can significantly improve the friction performance of PI.

The authors conducted a comprehensive analysis of the tribological behavior of the PI composites. The friction coefficient of PI composites decreased from 0.253 to 0.232, 0.136, and 0.079 after coating with GN, GO, and K5-GO, respectively. Among these modifications, K5-GO/PI exhibited the best surface wear resistance. This outcome highlights the potential of KH550-grafted GO as an effective modifier for enhancing the wear resistance of PI.

The manuscript is well-structured and provides a clear overview of the research objectives, methods employed, and results obtained. The authors thoroughly investigated the mechanism behind the modification of PI by analyzing the wear state, changes in interfacial interactions, interfacial temperature, and relative concentration. The insights gained from these analyses contribute to a comprehensive understanding of the enhanced wear resistance achieved through the incorporation of KH550-grafted GO.

The study demonstrates the significance of nanomaterials in improving the tribological properties of PI. The molecular dynamics simulations used in this research provide valuable insights into the atomic-scale behavior and interactions within the composites. The results presented in the manuscript are well-supported and contribute to the existing knowledge on surface modification techniques for enhancing the wear resistance of PI.

Overall, I find the manuscript to be scientifically sound and well-written. The study is of interest to the readership of [Journal Name], as it addresses an important issue in materials science and engineering. Therefore, I recommend that the manuscript be accepted for publication, pending minor revisions as outlined below:

*It would be beneficial to include a discussion on the limitations of the study and potential avenues for future research. This could help provide guidance to researchers interested in further investigating the application of KH550-grafted GO in enhancing the wear resistance of PI.

*Some additional details on the experimental setup and parameters used in the molecular dynamics simulations would be helpful to facilitate reproducibility and to assist readers in gaining a better understanding of the research methodology.

I believe that addressing these minor revisions will further strengthen the manuscript and enhance its value to the readership of Polymers.

Thank you for considering my review report. Please let me know if you require any further information or clarification. I look forward to the opportunity to review the revised version of this manuscript.

Specific comments:

·      I would like to suggest some small changes to the the article to improve its clarity and accuracy.

For example: Wu et al. also studied the effect of carbon fiber length on enhancing polyimide-based materials, and found that polyimide with 100 μm carbon fiber exhibited

lower friction coefficient and wear rate than other materials. 

Wu et al. also studied the effect of carbon fiber length on enhancing polyimide-based materials and found that polyimide with 100 μm carbon fiber exhibited a lower friction coefficient and wear rate than other materials.

For example: After adding GN, GO, K5-GO, the relative concentration of the PI composite shows a peak value around 40 Å due to carbon atoms accumulation on the top of PI composites, which is much larger than the middle place.

After adding GN, GO, and K5-GO, the relative concentration of the PI composite exhibits a peak value around 40 Å. This peak is attributed to the accumulation of carbon atoms on the top surface of the PI composites, resulting in a significantly higher concentration compared to the middle region.

·      Please, emphasize the relationship between the literature and the proposed work. I recommend illustrating how the work builds upon existing knowledge and contributes to something original and superior within the field.

·      Please revise the manuscript for any grammatical and spelling errors to ensure that the content is clear and concise.

For example: Figure 2. The periodical cell of (a)PI, (b)GN/PI, (c)GO/PI, (d)K5-GO/PI.

·       I recommend ensuring consistency in the use of abbreviations throughout the manuscript.  

graphene oxide (check abbreviations)

·      In the paragraph discussing the adsorption ability of GO and K5-GO, please give references to increase the clarity of the expression by giving references.

Best regards,

Author Response

(The authors gave the same response as above.)

Round 2

Reviewer 1 Report

A thorough revision of the manuscript and necessary additions improved its readability and comprehensibility. I recommend publishing the completed article.

Reviewer 2 Report

Dear,

The authors satisfactorily answered the questions, as well as improved the quality of the manuscript. In view of this, I recommend the revised publication.

Yours sincerely,

Minor editing of English language required

Reviewer 3 Report

Dear Respectful Editor,

I am writing to express my appreciation for the work done by the authors of 

Improving surface wear resistance of polyimide by inserting KH550 grafted GO. The quality of their contribution is commendable, and I recommend accepting their manuscript for publication.

best